# Anticoagulation Prior to COVID-19 Infection Has No Impact on 6 Months Mortality: A Propensity Score–Matched Cohort Study

**DOI:** 10.3390/jcm11020352

**Published:** 2022-01-12

**Authors:** Marcin Protasiewicz, Konrad Reszka, Wojciech Kosowski, Barbara Adamik, Wojciech Bombala, Adrian Doroszko, Damian Gajecki, Jakub Gawryś, Maciej Guziński, Maria Jedrzejczyk, Krzysztof Kaliszewski, Katarzyna Kilis-Pstrusinska, Bogusława Konopska, Agnieszka Kopec, Krzysztof Kujawa, Anna Langner, Anna Larysz, Weronika Lis, Lilla Pawlik-Sobecka, Joanna Gorka-Dynysiewicz, Marta Rosiek-Biegus, Agnieszka Matera-Witkiewicz, Tomasz Matys, Michał Pomorski, Mateusz Sokolski, Janusz Sokołowski, Anna Tomasiewicz-Zapolska, Katarzyna Madziarska, Ewa A Jankowska

**Affiliations:** 1Institute of Heart Diseases, Wroclaw Medical University, Borowska 213, 50-556 Wroclaw, Poland; marcin.protasiewicz@umw.edu.pl (M.P.); wojciech.kosowski@umw.edu.pl (W.K.); alangner-hetman@usk.wroc.pl (A.L.); mateusz.sokolski@umw.edu.pl (M.S.); 2Institute of Heart Diseases, University Hospital, 50-367 Wroclaw, Poland; anna.larysz@umw.edu.pl (A.L.); wlis@usk.wroc.pl (W.L.); atomasiewicz-zapolska@usk.wroc.pl (A.T.-Z.); ewa.jankowska@umw.edu.pl (E.A.J.); 3Department of Anesthesiology and Intensive Therapy, Wroclaw Medical University, 50-367 Wroclaw, Poland; barbara.adamik@umw.edu.pl; 4Statistical Analysis Centre, Wroclaw Medical University, 50-367 Wroclaw, Poland; wojciech.bombala@umw.edu.pl (W.B.); krzysztof.kujawa@umed.edu.pl (K.K.); 5Clinical Department of Internal and Occupational Diseases, Hypertension and Clinical Oncology, Wroclaw Medical University, 50-367 Wroclaw, Poland; adrian.doroszko@umw.edu.pl (A.D.); damian.gajecki@umed.wroc.pl (D.G.); jakub.gawrys@umed.wroc.pl (J.G.); tomasz.matys@umw.edu.pl (T.M.); 6Department of Radiology, Wroclaw Medical University, 50-367 Wroclaw, Poland; maciej.guzinski@umw.edu.pl; 7Department of Nursing and Obstetrics, Division of Internal Medicine Nursing, Wroclaw Medical University, 50-367 Wroclaw, Poland; maria.jedrzejczyk@umw.edu.pl; 8Clinical Department of General, Minimally Invasive and Endocrine Surgery, Wroclaw Medical University, 50-367 Wroclaw, Poland; krzysztof.kaliszewski@umw.edu.pl; 9Clinical Department of Paediatric Nephrology, Wroclaw Medical University, 50-367 Wroclaw, Poland; katarzyna.kilis-pstrusinska@umw.edu.pl; 10Department of Pharmaceutical Biochemistry, Division of Pharmaceutical Biochemistry, Wroclaw Medical University, 50-367 Wroclaw, Poland; boguslawa.konopska@umed.wroc.pl (B.K.); joanna.gorka-dynysiewicz@umw.edu.pl (J.G.-D.); 11Clinical Department of Internal Medicine, Pneumology and Allergology, Wroclaw Medical University, 50-367 Wroclaw, Poland; agnieszka.kopec@umw.edu.pl (A.K.); marta.rosiek-biegus@umw.edu.pl (M.R.-B.); 12Clinical Department of Heart Transplantation and Mechanical Circulatory Support, Institute of Heart Disease, Wroclaw Medical University, 50-367 Wroclaw, Poland; 13Division of Basic Sciences, Faculty of Health Sciences, Wroclaw Medical University, 50-367 Wroclaw, Poland; lilla.pawlik-sobecka@umw.edu.pl; 14Screening of Biological Activity Assays and Collection of Biological Material Lab., Wroclaw Medical University Biobank, Wroclaw Medical University, 50-367 Wroclaw, Poland; agnieszka.matera-witkiewicz@umw.edu.pl; 152nd Department of Gynaecology and Obstetrics, Wroclaw Medical University, 50-367 Wroclaw, Poland; michal.pomorski@umw.edu.pl; 16Clinical Department of Emergency Medicine, Wroclaw Medical University, 50-367 Wroclaw, Poland; janusz.sokolowski@umw.edu.pl; 17Department of Nephrology and Transplantation Medicine, Wroclaw Medical University, 50-367 Wroclaw, Poland; katarzyna.madziarska@umw.edu.pl; 18Department of Translational Cardiology and Clinical Registries, Institute of Heart Disease, Wroclaw Medical University, 50-367 Wroclaw, Poland

**Keywords:** COVID-19, anticoagulation, thromboembolic complications, SARS-CoV-2, morbidity, mortality, 6 months outcome

## Abstract

The coronavirus disease 2019 (COVID-19) shows high incidence of thromboembolic events in humans. In the present study, we aimed to evaluate if anticoagulation prior to COVID-19 infection may impact clinical profile, as well as mortality rate among patients hospitalized with COVID-19. The study was based on retrospective analysis of medical records of patients with laboratory confirmed SARS-CoV-2 infection. After propensity score matching (PSM), a group of 236 patients receiving any anticoagulant treatment prior to COVID-19 infection (AT group) was compared to 236 patients without previous anticoagulation (no AT group). In 180 days, the observation we noted comparable mortality rate in AT and no AT groups (38.5% vs. 41.1%, *p* = 0.51). Similarly, we did not observe any statistically significant differences in admission in the intensive care unit (14.1% vs. 9.6%, *p* = 0.20), intubation and mechanical ventilation (15.0% vs. 11.6%, *p* = 0.38), catecholamines usage (14.3% vs. 13.8%, *p* = 0.86), and bleeding rate (6.3% vs. 8.9%, *p* = 0.37) in both groups. Our results suggest that antithrombotic treatment prior to COVID-19 infection is unlikely to be protective for morbidity and mortality in patients hospitalized with COVID-19.

## 1. Introduction

On March 11, 2020, the World Health Organization (WHO) declared the coronavirus disease of 2019 (COVID -19), caused by “severe acute respiratory syndrome coronavirus 2” (SARS-CoV-2), as a pandemic [1]. As of September 2021, COVID-19 has killed almost 4.5 million people worldwide, while more than 200 million people have been infected and those numbers are growing [2].

SARS-CoV-2 is a single-stranded RNA virus, first identified in the city of Wuhan located in Hubei, China [3,4]. While most of the SARS-CoV-2 infections have a mild to moderate course, about 21% of all cases have severe and critical presentation [4]. The infection fatality rates vary from 0.3 to 5%, and the leading causes of death are respiratory failure, sepsis/multi-organ failure, cardiac failure, hemorrhage, and renal failure [5,6,7]. Furthermore, COVID-19 is associated with hypercoagulability and thromboembolic complications that can be fatal even in asymptomatic and mild infection [8,9]. The risk of mortality associated with thromboembolism is however the highest in the patients treated in the ICU [10].

Recent studies show various incidence rates of thromboembolic events, including venous and arterial thromboembolism (4.4–85.4% and 2.8–8.4%, respectively), pulmonary embolism (up to 87%), stroke (2.5%), acute coronary syndromes (1.1%), as well as bowel and limb ischemia (0.7 and 0.66%, respectively) [11]. Moreover, almost 70% of patients with severe SARS-CoV-2 infection present features of disseminated intravascular coagulation (DIC) [12]. Microthrombosis may be even more frequent-post-mortem evaluation revealing small fibrinous thrombi in small pulmonary arterioles in 80% of COVID-19 fatal cases [13].

SARS-CoV-2 infection can cause thrombosis by several mechanisms at the molecular and cellular level, including dysregulation of the renin-angiotensin-aldosterone system (RAAS) and immune response [14,15]. The virus enters human cells through binding to the human angiotensin-converting enzyme 2 (ACE2) receptor, which is expressed in cells of nasopharynx and lungs, as well as endothelia of blood vessels (especially coronary and renal blood vessels) and cells of heart, kidney, testicle, and brain [16,17,18]. By inactivating angiotensin II, ACE2 plays a major role as a negative regulator of RAAS [19]. Downregulation of ACE2 associated with COVID-19 leads to an increase of angiotensin II, a proinflammatory and prothrombotic peptide [20,21].

Hypercoagulability in COVID-19 is also due to dysregulation of the immune response [14]. Elevated levels of acute phase reactants, like factor VIII, von Willebrand factor, and fibrinogen increase risk of blood coagulation. Moreover, the uncontrolled activation of innate immunity pathways (called ‘immunothrombosis’) leads to microthrombus formation and exacerbation of inflammation [22,23]. Severe course of the disease if often associated with an increase of inflammatory cytokines, especially interleukin 6 (IL-6), which can eventually result in cytokine storm, a life-threatening inflammatory syndrome [23,24] contributing to hypercoagulable state and endothelial dysfunction thus increasing the risk of disseminated intravascular coagulation [25,26,27,28,29].

Consequently, there is a need for effective anticoagulation in groups of patients diagnosed with COVID-19, in order to reduce incidence of thromboembolic complications. However, the anticoagulants of choice, optimal doses, and the groups of patients that will especially benefit from this treatment are still to be determined. Moreover, there is limited data on the prognosis of patients requiring oral anticoagulation (OAC) prior to the COVID-19 infection.

In the present study, our aim was to evaluate the effect of prior to COVID-19 infection anticoagulation treatment on clinical profile, incidence of thromboembolic and hemorrhagic complications as well as middle-term mortality among patients hospitalized with COVID-19.

## 2. Materials and Methods

### 2.1. Study Design and Participants

We analyzed retrospectively 2070 medical records of patients with laboratory confirmed SARS-CoV-2 infection, who were admitted and hospitalized at the Medical University Hospital in Wroclaw (Poland) between March 2020 and May 2021. Individuals under 18 were excluded from the study.

Infection with COVID-19 was confirmed by a positive result on a reverse transcriptase polymerase chain reaction severe acute respiratory syndrome coronavirus 2 assay. All patients were divided into two groups according to antithrombotic treatment status. Patients receiving any anticoagulant treatment (low molecular weight heparin (LMWH), vitamin K antagonists (VKAs) or direct-acting oral anticoagulants (DOACs) prior to COVID-19 infection (AT group) were compared to patients without previous anticoagulation (no AT group).

The study protocol was approved by the Institutional Review Board and Ethics Committee of Wroclaw Medical University, Wroclaw, Poland (No: KB-444/2021). The routine data were collected retrospectively; therefore, written informed consent to participate in the study was not required. The Bioethics Committee approved the publication of anonymized data.

### 2.2. Data Sources

Data were collected from electronic medical records. Analyzed variables included demographics, laboratory measurements, comorbidities, and outcomes.

### 2.3. Outcomes

The primary outcome was 180 days all-cause mortality. We analyzed also rate of significant adverse events like the need for intubation and invasive mechanical ventilation, need for catecholamines usage, and major (World Health Organization grade ≥3) bleeding. Laboratory parameters were considered elevated as defined by local laboratory cut-off levels. Major bleeding was defined using International Classification of Diseases-10th Revision codes or receiving ≥2 packed red blood cell transfusions.

### 2.4. Statistics

Descriptive data were presented as numbers and percentages for categorical variables, and as the mean, standard deviation, median, and interquartile range (IQ) for numerical variables. The distribution of continuous variables was tested using Kolmogorov–Smirnov/Shapiro–Wilk tests. The Chi-square test or Fisher exact test were used for the comparison of qualitative variables. The Mann–Whitney U test was used for subgroup analysis of non-normally distributed variables and Student’s t-test was used for the comparison of means for normally distributed data. In multiple group comparisons of numerical variables, the Kruskal–Wallis test was used for non-normally distributed numerical variables.

To compare the risk of above defined outcomes among patients with or without prior antithrombotic therapy we conducted a propensity score matching (PSM). Propensity scores were calculated using a logistic regression model, adjusting for the following covariates: age, sex, hypertension, heart failure, previous stroke, renal insufficiency, obesity, diabetes. Patients were matched 1:1 across each cohort on a propensity score generated by logistic regressions using the nearest neighbor technique. As a result, pairs of patients balanced in respect to variables which could influence the outcome were selected from the entire population of 2070 patients.

Survival analyses by Kaplan-Meier estimates were performed to assess differences in event-free (primary outcome) survival between AT and no AT patients before and after PSM. 

All the statistical tests were two-tailed and the statistical significance level was set at *p* < 0.05. The analyses were performed using Statistica v.13.3 (TIBCO Software Inc., Palo Alto, CA, USA) except PSM which was performed with the use of R-package MatchIt [30].

## 3. Results

The entire study group consisted of 2070 COVID-19 patients. The clinical characteristics of the entire study population are presented in Table 1. There were 296 (14.2%) patients receiving anticoagulation prior to COVID -19 infection. Among them 146 (7%) patients receiving LMWH, 104 (5%) receiving DOACs, and 46 (2.2%) on VKA treatment.

During hospitalization 9.8% of all patients required transfer to the intensive care unit, 9.7% catecholamines usage, and 9.7% needed intubation and mechanical ventilation. Bleeding complications were observed in 5.2% of patients. In 180 days follow-up observation, a 37.9% mortality rate was noted in the entire population.

AT patients had a significantly worse 180 days prognosis when compared to no AT group (mortality rate 52.0% vs. 35.9%, respectively, *p* < 0.001). Kaplan–Meier analysis (log-rank test) also confirmed significantly lower 180 days survival in AT group (Figure 1). However, baseline characteristics of COVID-19 patients with prior AT were significantly different as compared to the no AT group in terms of important comorbidities. For this reason, we performed a PSM (1:1) analysis to balance both cohorts in respect to variables which could influence the outcome.

After PSM, the group of 236 patients receiving AT before hospitalization and 236 patients without previous anticoagulation were selected from the whole study population. The characteristics of two groups of patients (236:236) after PSM are shown in Table 2 and Table 3.

After PSM we noted that similar rates of patients in on- and off treatment groups required admission in the intensive care unit (ICU) (14.1% vs. 9.6%, respectively, *p* = 0.20), intubation and mechanical ventilation (15.0% vs. 11.6%, respectively, *p* = 0.38) or catecholamines usage (14.3% vs. 13.8%, respectively, *p* = 0.86). Similarly, bleeding rate was comparable in both groups (6.3% vs. 8.9%, respectively, *p* = 0.37). 

Regarding the primary outcome of the study, 38.5% of AT patients after PSM died during the 180 days follow-up. This was comparable to the 41.1% rate of death among no AT patients after PSM (*p* = 0.51). Kaplan–Meier analysis (log-rank test) also demonstrated similar survival in both groups of patients until the 180^th^ day from admission to hospital (Figure 2).

When comparing effects of different anticoagulation management protocols on the primary outcome we noted no significant differences between patients on LWMH, VKA, or DOACs therapy (Figure 3).

We did not observe significant differences in laboratory parameters at admission between matched populations of AT and no AT patients. Measurements concerned hemoglobin, lymphocytes, leukocytes, platelets, total protein, albumin CRP, procalcitonin, and IL-6, fibrinogen (Table 3).

## 4. Discussion

Our study demonstrated that anticoagulation used prior to COVID-19 has no impact on middle-term mortality. What is more, no method of anticoagulation is superior to any other. Moreover, prior antithrombotic therapy had no impact on rate of intubation and mechanical ventilation, catecholamines usage or bleeding rate in 180 days follow up or laboratory parameters during admission to the hospital.

Nadkarni et al. had demonstrated that anticoagulation is associated with lower mortality and intubation among hospitalized COVID-19 patients [31]. There were no statistically significant differences in mortality reduction between prophylactic and therapeutic regimens [31]. However, in this study anticoagulation was initiated within 48 h from admission, and the impact of previous anticoagulation on survival was not analyzed [31]. What is more, individual agents were only compared in descriptive analyses, thus no conclusions can be drawn.

Interestingly, results of the HEP-COVID randomized clinical trial showed reduction of mortality and thromboembolic events among patients with COVID-19 treated with therapeutic-dose LMWH compared with standard heparin thromboprophylaxis [32]. However, the study group included only patients with high-risk of thromboembolism (defined D-dimer levels more than 4 times the upper limit of normal or sepsis-induced coagulopathy score of 4 or greater). Thus, results of these study may not be generalizable to hospitalized patients who are less acutely ill [32]. HEP-COVID study results suggest that thromboprophylaxis with therapeutic-dose low-molecular-weight heparin may reduce thromboembolism and death in high-risk inpatients but who are not in critical condition as the treatment effect was not seen in ICU patients [32].

This hypothesis is consistent with two recent randomized studies. Lawler et al. proved that in noncritically ill patients, therapeutic-dose anticoagulation with heparin increased the probability of survival to hospital discharge with reduced use of cardiovascular or respiratory organ support as compared with usual-care thromboprophylaxis [33]. However, Golinger et al. reported that in critically ill patients with COVID-19, an initial strategy of therapeutic-dose anticoagulation with heparin did not result in a greater probability of survival to hospital discharge or a greater number of days free of cardiovascular or respiratory organ support than did usual-care thromboprophylaxis [34].

In the meta-analysis of 14 studies involving 7681 patients hospitalized with SARS-CoV-2 infection, preventive and therapeutic anticoagulation had significant effectiveness in reducing mortality risk [33]. Yet, the odds of developing deep venous thrombosis and pulmonary embolism were not statistically significant between the cohorts treated with no anticoagulation and prophylactic anticoagulation and was not statistically significant in the comparison between no anticoagulation and anticoagulation [35]. 

The results of our paper are however consistent with Tremblay et al. observations. Authors analyzed the association of anticoagulation (i.e., VKAs, DOACs, or heparins) on the regular basis prior to SARS-CoV-2 infection with the outcomes of COVID-19 and reported that it brought no benefit in preventing severe forms of COVID-19 and in reducing all-cause mortality [36].

Thromboembolic events are considered an important complication among patients with COVID-19. Hypercoagulability connected with SARS-CoV-2 infection and consecutive complications urges for effective and safe anticoagulation. Nevertheless, some patients require chronic anticoagulation, mostly due to atrial fibrillation and venous thromboembolism. These patients are usually elderly, with multiple underlying medical conditions increasing their risk of complications and mortality after SARS-CoV-2 infection [37,38,39]. In the meta-analysis of 187,716 adults, Romiti et al. reported that occurrence of atrial fibrillation (AF) during COVID-19 is associated with increased mortality [38]. Patients with AF and COVID-19 were more likely to be older, hypertensive, diabetic, with concomitant coronary artery disease or chronic heart failure, and in a critical clinical status. Moreover, there was a four-fold higher risk of death in the AF group compared to patients without AF [40]. The potential benefits of anticoagulation may not outweigh increased risk. Rivera-Caravaca et al. reported lower survival and higher mortality risk among COVID-19 patients on OAC therapy at hospital admission [37]. We observed also increased mortality among AT patients in the entire population; however, after PSM this effect was no longer observed.

Heparins (unfractionated heparin, UFH and LMWH) are well-known anticoagulants with proven anti-inflammatory activity. They reduce endothelial cell damage and inhibit adhesion of leukocytes to endothelium. What is more, heparin effect on IL-6 and IL-8 reduction has been demonstrated [14,41]. Recently, the role of heparin in reducing SARS-CoV-2 infectivity has been postulated. In vitro studies revealed a high binding affinity of heparin to the spike protein of SARS-CoV-2. As a result, heparin can directly inhibit spike protein binding to the ACE2 receptor [42,43]. While there is a strong pharmacologic rationale for heparins in treating SARS-CoV-2 infection, pre-hospital use of LMWH has not increased the middle-term survival of patients with COVID-19 in our study. However, it should be noted that the mortality rate noted in this group was similar to no AT patients and tended to be lower when compared to patients on VKAs or DOACs. 

In our study, out of 1774 patients who did not require anticoagulation before COVID-19, 1204 (67.8%) patients received LMWH during hospitalization. Anticoagulation with LMWH were used due to immobilization and hypercoagulability associated with SARS-CoV-2 infection. Other methods of anticoagulation were not used in this group.

The benefits of VKAs therapy and its impact on outcomes of patients hospitalized with COVID-19 remains controversial, as the results of available studies are inconclusive or even conflicting [31,37,44,45]. Some studies revealed decrease in vitamin K levels among patients with COVID-19 compared to uninfected controls and negative relationship between vitamin K concentration and the clinical severity of COVID-19 [12,46]. Therefore, it was hypothesized that VKAs therapy may be associated with reduced survival of patients with SARS-CoV-2 infection [47]. It was suggested that excessive decrease in vitamin K may paradoxically lead to thrombogenicity and exacerbation of lung fibrosis [47,48,49,50]. Moreover, the under-carboxylation of matrix Gla protein (MGP) in the absence of vitamin K is associated with accelerated arterial calcification [51,52].

Anastasi et al. reported that vitamin K deficiency in male COVID-19 patients was associated with greater IL-6 levels in the general circulation and consequently, cytokine storm and related fatal outcomes [47,53] However, results of available studies are inconsistent [47,54]. International Society of Thrombosis and Haemostasis (ISTH) Clinical Guidance on the Diagnosis, Prevention, and Treatment of Venous Thromboembolism in Hospitalized Patients with COVID-19 does not include guidelines for VKAs therapy among patients hospitalized with COVID-19 [55]. In our population, mortality rate in VKA treated patients did not differ significantly as compared to other AT patients; however, it reached the highest rate among all.

While DOACs can be used for in-hospital prophylaxis, they should be administered with caution due to possible interferences with anti-retroviral drugs, immunosuppressants, and other experimental therapies [53,54,56]. Moreover, in patients with unstable kidney function DOACs are likely to accumulate, thus monitoring of anticoagulation may be required [57,58]. We are waiting for the ongoing PREVENT-HD trials results. This study evaluates whether rivaroxaban reduces the risk of thrombotic events, all-cause hospitalization, and all-cause mortality in patients with acute COVID-19 infection [59]. The ACTION trial, a randomized, open-label, controlled, multicenter study investigated the outcomes of full anticoagulation using oral anticoagulants compared to prophylactic anticoagulation for hospitalized COVID-19 patients [60]. The data has revealed that full anticoagulation with rivaroxaban did not result in improved outcomes when compared to prophylactic anticoagulation. Our results show no benefit in outcome in DOACs patients group compared to other AT patients or to no AT group.

In conclusion, our results suggest that antithrombotic treatment prior to COVID-19 infection is unlikely to be protective for morbidity and mortality. Nevertheless, further prospective controlled trials are needed to validate the results of available observational studies and determine guidelines of anticoagulation among patients with COVID-19.

There are several important limitations of our study, including its retrospective character. Thus, results reported in this manuscript should be regarded only as hypothesis-generating. Another important limitation is the low power due to reduced sample size. Thus, the lack of significant differences between analyzed groups may be attributed to the low power rather than to a true lack of difference between the groups. We included a number of factors in our propensity score matching; however, there are others not included that could possibly impact analyzed outcomes. The high age of the included patients may contribute to a worse prognosis and high mortality. We also did not examine the influence of interventions after hospitalization which could also influence results. Regarding LMWH, it was impossible to be determined in all patients where the dose was prophylactic or therapeutic anticoagulant. As such, we decided not to include this information.

## Figures and Tables

**Figure 1 jcm-11-00352-f001:**
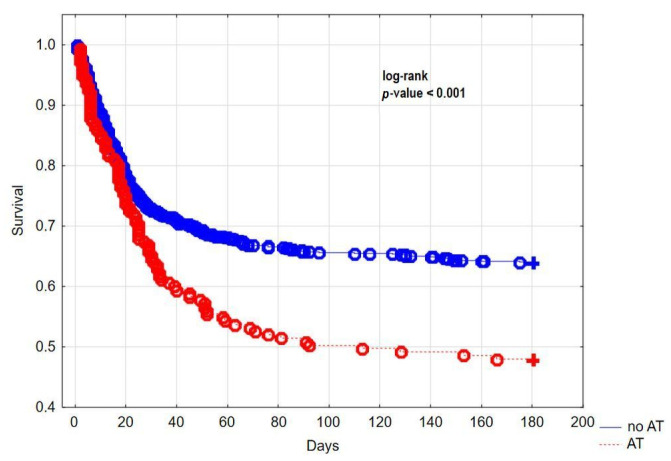
Kaplan–Meier curve showing 180 days survival stratified for patients with and without AT.

**Figure 2 jcm-11-00352-f002:**
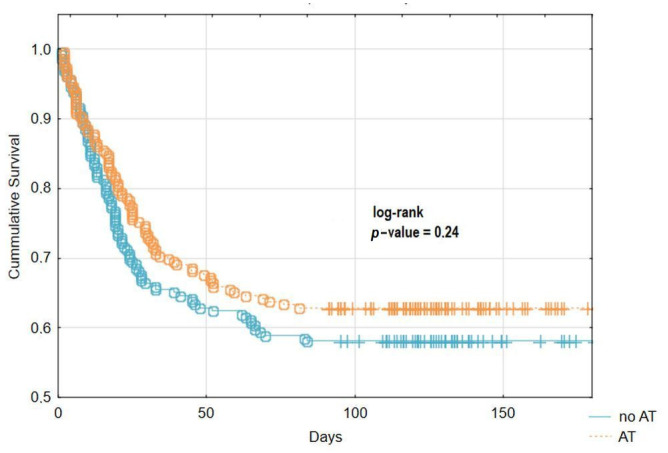
Kaplan–Meier curve showing cumulative 180 days survival stratified for patients with and without AT after PSM.

**Figure 3 jcm-11-00352-f003:**
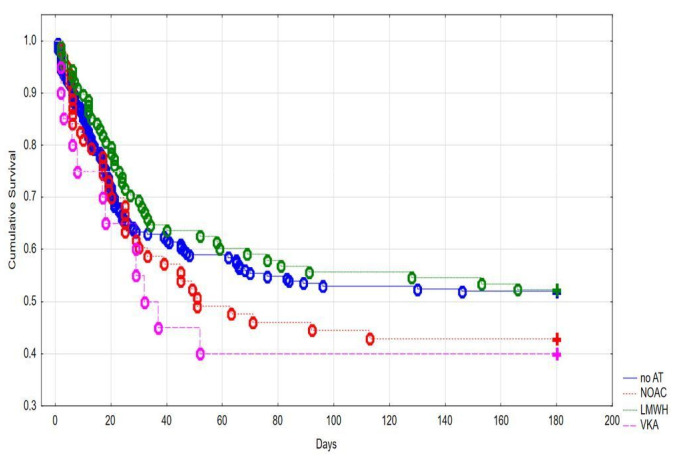
Kaplan–Meier curve showing cumulative survival stratified for patients with different anticoagulation management protocols.

**Table 1 jcm-11-00352-t001:** Baseline characteristics in total population.

Variables	Overall Population *N* = 2070
Age (years), median (IQR)	63.0 (45–73)
Male sex, *n* (%)	1043 (50.4)
Obesity *, *n* (%)	513 (24.8)
Hypertension, *n* (%)	947 (45.7)
Diabetes mellitus, *n* (%)	472 (22.8)
Heart failure, *n* (%)	191 (9.2)
Stroke, *n* (%)	145 (7.0)
Chronic kidney disease, *n* (%)	192 (9.3)
Hypercholesterolaemia, *n* (%)	1035 (50)
Smoking, *n* (%)	177 (8.5)
History of malignant disease, *n* (%)	140 (6.7)

* BMI > 30 kg/m^2^.

**Table 2 jcm-11-00352-t002:** Baseline characteristics in matched patients’ populations.

Variables	No AT *N* = 236	AT *N* = 236	*p*-Value
Age (years), median (IQR)	71 (64–78)	72 (64–83)	*p* = 0.23
Male sex, *n* (%)	124 (52.5)	128 (54.2)	*p* = 0.71
Obesity *, *n* (%)	78 (33.0)	69 (29.2)	*p* = 0.38
Diabetes mellitus, *n* (%)	87 (36.9)	86 (36.4)	*p* = 0.92
Hypertension, *n* (%)	61 (25.8)	68 (28.8)	*p* = 0.46
Stroke, *n* (%)	46 (19.5)	44 (18.6)	*p* = 0.81
Chronic kidney disease, *n* (%)	46 (19.5)	47 (19.9)	*p* = 0.90
Hypercholesterolaemia, *n* (%)	104 (44.0)	94 (39.8)	*p* = 0.47
Smoking, *n* (%)	20 (8.5)	21 (8.9)	*p* = 0.30
Heart failure, *n* (%)	44 (18.6)	51 (21.6)	*p* = 0.42
History of malignant disease, *n* (%)	0 (0)	1 (0.4)	*p* = 0.36
Treatment before hospitalization			
Angiotensin-converting enzyme antagonists, *n* (%)	46 (19.5)	87 (36.8)	*p* < 0.005
β-blockers, *n* (%)	52 (22.0)	45 (19.0)	*p* = 0.42
Angiotensin receptor blockers, *n* (%)	24 (10.1)	19 (8.0)	*p* = 0.42
Mineralocorticoid receptor antagonists, *n* (%)	24 (10.1)	17 (7.2)	*p* = 0.25
Calcium channel blockers, *n* (%)	41 (17.4)	65 (24.5)	*p* < 0.01
Loop diuretics, *n* (%)	17 (7.2)	24 (10.1)	*p* = 0.25
Thiazide or thiazide-likediuretics, *n* (%)	77 (32.6)	49 (20.7)	*p* < 0.005
Insulin, *n* (%)	22 (9.3)	25 (10.6)	*p* = 0.64
Statins, *n* (%)	127	97	*p* = 0.06

No AT—patients without antithrombotic treatment; AT—patients with antithrombotic treatment; * BMI >30 kg/m^2^.

**Table 3 jcm-11-00352-t003:** Laboratory values in matched patients populations.

	No AT *N* = 236	AT *N* = 236	*p*-Value
HGB, g/dL	12.60 (3.10)	12.35 (3.50)	*p* = 0.34
WBC, 103/µL	8.29 (4.78)	8.06 (6.13)	*p* = 0.43
Lymphocytes, 103/µL	1.02 (0.78)	0.98 (0.81)	*p* = 0.49
PLT, 103/µL	225.00 (127.00)	202.50 (129.00)	*p* = 0.19
CRP, mg/dL	59.21 (92.94)	69.19 (108.82)	*p* = 0,55
Fibrinogen, g/L	4.76 (2.30)	4.60 (2.64)	*p* = 0.28
IL-6, pg/mL	19.36 (34.22)	23.34 (46.23)	*p* = 0.42
PCT ng/mL	0.52 (1.32)	0.69 (1.02)	*p* = 0.39
Creatinine, mg/dL	1.11 (0.64)	1.20 (0.91)	*p* = 0.14
Albumin, g/dL	3.20 (0.80)	3.10 (0.80)	*p* = 0.67
Total protein, g/dL	6.00 (1.10)	5.90 (1.10)	*p* = 0.59

No AT—patients without antithrombotic treatment; AT—patients with antithrombotic treatment; HGB: hemoglobin; WBC: white blood cells; PLT: platelets; CRP: C reactive protein; IL-6: interleukin-6; PCT: procalcitonin; Data are given as median and interquartile range.

## Data Availability

The datasets used and/or analyzed during the current study are available from the corresponding author upon reasonable request.

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
