# Peer review of "Anticoagulation Prior to COVID-19 Infection Has No Impact on 6 Months Mortality: A Propensity Score–Matched Cohort Study"

_jcm, 2022, doi:10.3390/jcm11020352_

Round 1

Reviewer 1 Report

Interesting article. Any doubts:
Data from 472 patients are analyzed when 2,070 patients were selected, what happened here? The methodology is not well explained, as well as the distribution of patients with anticoagulation or not.

Reviewer 2 Report

Protasiewicz et al. presented an elegant manuscript on the outcomes among patients affected by COVID-19 in two different group: patients receiving anticoagulant treatment (AT) prior to COVID-19; and patients receiving no AT prior to COVID-19.

The strenght of this paper lies in the propensity score match (PSM) analysis which reinforces the data presented. Seems resonable that patients receiving anticoagulants had more comorbidities compared to the ones did not receive anticoagulant, that could be particularly evident in the first analysis in which AT group has worst outcomes; however, when analysing in PSM, similar rate in ICU admission, ventilation or cathecolamin usage were reported, as well as for 6-months mortality.

Did you consider the use of anticoagulant during hospitalization for COVID-19? I think you should report this information if available. I suggest to comment this data (or the lack of this data) on the discussion and eventually, adjust for this counfounder the analysis presented.

Do you may want to discuss your result in the light of a previous Meta-analysis in which patients with AF and COVID-19 had worst outcomes (doi:10.3390/jcm10112490). 
